# Recommendations of high-quality clinical practice guidelines related to the process of starting dialysis: A systematic review

Karla Salas-Gama[1,2,3]*, Igho J. Onakpoya[4], Jorge Coronado Daza[5], Rafael Perera[4], Carl J. Heneghan[4]

1 Quality, Process and Innovation Direction, Hospital Universitari Vall d'Hebron, Barcelona, Spain, 2 Health Services Research Group, Vall d'Hebron Institut de Recerca (VHIR), Vall d'Hebron Hospital Universitari, Barcelona, Spain, 3 Consortium for Biomedical Research in Epidemiology and Public Health–CIBERESP, Barcelona, Spain, 4 Centre for Evidence-Based Medicine, Nuffield Department of Primary Care Health Sciences, University of Oxford, Oxford, United Kingdom, 5 Medicine Department, Universidad de Cartagena, Cartagena, Bolívar, Colombia

* ksalas@vhebron.net

**Data Availability Statement:** All relevant data are within the paper and its Supporting Information files.

## Abstract

### Background

The optimal time for initiation of dialysis and which modality to choose as the starting therapy is currently unclear. This systematic review aimed to assess the recommendations across high-quality clinical practice guidelines (CPGs) related to the start of dialysis.

### Methods

We systematically searched MEDLINE, EMBASE, Web of Science, LILACS, and databases of organisations that develop CPGs between September 2008 to August 2021 for CPGs that addressed recommendations on the timing of initiation of dialysis, selection of dialysis modality, and interventions to support the decision-making process to select a dialysis modality. We used the Appraisal of Guidelines for Research and Evaluation instrument to assess the methodological quality of the CPGs and included only high-quality CPGs. This study is registered in PROSPERO, number CRD42018110325.

### Results

We included 12 high-quality CPGs. Six CPGs addressed recommendations related to the timing of initiating dialysis, and all agreed on starting dialysis in the presence of symptoms or signs. Six CPGs addressed recommendations related to the selection of modality but varied greatly in their content. Nine CPGs addressed recommendations related to interventions to support the decision-making process. Eight CPGs agreed on recommended educational programs that include information about dialysis options. One CPG considered using patient decision aids a strong recommendation.

**Funding:** The author(s) received no specific funding for this work.

**Competing interests:** IJO and CJH have held grant funding from the NIHR Evidence Synthesis Working Group (ESWG Grant no: 390). CJH has received expenses and fees for his media work (including payments from BBC Radio 4 Inside Health). He has received expenses from the WHO, FDA, and holds grant funding from the NIHR, the NIHR School of Primary Care Research, the NIHR BRC Oxford and the WHO. He has received financial remuneration from an asbestos case and given free legal advice on mesh cases. He has also received income from the publication of a series of toolkit books published by Blackwells. On occasion, he receives expenses for teaching EBM and is also paid for his GP work in NHS out of hours (contract with Oxford Health NHS Foundation Trust). RP hold grant funding from the NIHR Programme of Applied Research. He leads a programme looking at how general practitioners manage chronic kidney disease and chronic heart failure. This does not alter our adherence to PLOS ONE policies on sharing data and materials. KS and JC have no interests to declare.

**Abbreviations:** AGREE II, Appraisal of Guidelines for Research and Evaluation—II; CKD, chronic kidney disease; CPG(s), clinical practice guidelines; eGFR, estimated glomerular filtration rate; ESKD, end-stage kidney disease; ADPKD, autosomal dominant polycystic kidney disease; GRADE, Grades of Recommendation, Assessment, Development and Evaluation; HD, haemodialysis; NICE, National Institute for Health and Care Excellence; PD, peritoneal dialysis; PRISMA, Preferred Reporting Items for Systematic Reviews and Meta-Analysis; RRT, renal replacement therapy.

## Limitations

We could have missed potentially relevant guidelines since we limited our search to CPGs published from 2008, and we set up a cut-off point of 60% in domains of the rigour of development and editorial independence.

## Conclusion

High-quality CPGs related to the process of starting dialysis were consistent in initiating dialysis in the presence of symptoms or signs and offering patients education at the point of decision-making. There was variability in how CPGs addressed the issue of dialysis modality selection. CPGs should improve strategies on putting recommendations into practice and the quality of evidence to aid decision-making for patients.

## Registration

The protocol of this systematic review has been registered in the international prospective register of systematic reviews (PROSPERO) under the registration number: CRD CRD42018110325.

https://clinicaltrials.gov/ct2/show/CRD42018110325.

## Introduction

Chronic kidney disease (CKD) is a global health problem with more than one in ten adults affected [1]. Patients with end-stage kidney disease (ESKD) require renal replacement therapy (RRT), and most will do it with one of the two dialysis modalities: peritoneal dialysis (PD) or haemodialysis (HD). Starting dialysis is a complex decision, and the optimal time for starting is unclear. The only randomised controlled trial that analysed an early vs late-start showed no differences between the two approaches, and concluded that with careful clinical management, dialysis should be delayed until eGFR reaches 7ml/min/1.73m$^2$ or clinical symptoms are present [2].

There is currently insufficient evidence about which modality to choose as the starting therapy. In 2018, more than three million people worldwide were on dialysis. Of these, only 11% were on PD [3]. A randomised controlled trial compared starting dialysis with HD vs PD showed no differences in quality of life and mortality at two years [4]. Nevertheless, the quality of the study is difficult to determine, since the trial was reported in abstract form only, and it stopped recruitment before the prespecified sample size than intended due to poor recruitment. A previous Cochrane review also concluded there was insufficient data to draw conclusions [5]. Given the lack of good quality evidence to recommend one modality over the other and considering that HD and PD have different practical factors, including harms and benefits that will potentially impact a person's life, it is particularly important to offer patients an evidence-based, individualised decision-making process.

Clinical practice guidelines (CPGs) offer users clinical recommendations for daily practice based on the best available evidence. The Institute of Medicine defines CPGs as 'recommendations for clinicians about the care of patients with specific conditions. They should be based upon the best available research evidence and practice experience' [6]. However, clinicians are faced with guidelines of variable quality and not sufficiently transparent with respect to

principles for guideline development, evidence review, or potential conflicts of interest [7]. To be able to use CPGs in clinical practice, their quality needs to be ensured.

Several CPGs for the management of CKD have been published in different countries. However, no systematic assessment has been done on the process of starting dialysis recommendations. The objective of this systematic review was to assess the consistency across CPGs recommendations in three essential themes: timing of initiation of dialysis, selection of dialysis modality, and the interventions to support the decision-making process about dialysis modality selection.

## Methods

This study was reported in line with the Preferred Reporting Items for Systematic Reviews and Meta-Analysis (PRISMA) statement [8]. No amendments were made to the protocol after its registration other than an updated search.

### Inclusion and exclusion criteria

**Population.** We included high-quality CPGs related to the process of starting dialysis in adults (18 years or more), published in English or Spanish between January 2008 until the present. Where more than one version of the same guideline was found, we included only the most updated version. We defined high-quality CPGs as guidelines that were evidence-based (explicitly describing how the evidence was assessed) and that obtained a minimum score of 60% in domain 3 (Rigor of Development) and domain 6 (Editorial Independence) at the Appraisal of Guidelines for Research and Evaluation (AGREE II) instrument [9]. We considered these domains because the rigour of guideline development and the editorial independence of authors seem to have the strongest influence on the overall assessment of guideline quality and recommendation for use [10]. We established the cut-off of 60% before beginning the AGREE II appraisals based on values adopted by other authors [11–13].

**Interventions.** CPGs recommendations related to three specific themes of the process of starting dialysis: a) timing of initiation of dialysis (criteria to when to start dialysis treatment), b) dialysis modality selection (in-centre or home HD, or ambulatory or automated PD), and c) interventions to support the decision-making process about dialysis modality selection (any tool designed to help people participate in shared decision-making process, like educational programs, decision aids, algorithms, peer support programs, etc.).

**Outcomes.** a) Variation across the content of included high-quality CPG recommendations on the process of initiating dialysis in CKD patients; b) methodological quality of included guidelines.

We also collected data on organisation or author, year, country, language, target population, level of evidence and grade of recommendation.

**Exclusion criteria.** CPGs that did not offer recommendations related to the timing of initiation of dialysis treatment, dialysis modality selection, or interventions to support the decision-making process were excluded. CPGs focusing on acute kidney failure, pregnancy, or paediatric populations were excluded. Adaptations, translations, commentaries, or summaries were also excluded.

### Identification of clinical practice guidelines

CPGs were identified through a systematic search of MEDLINE, Web of Science, EMBASE, LILACS, and databases of organisations that develop CPGs like the National Institute for Health and Care Excellence (NICE), Scottish Intercollegiate Guidelines, and the Guidelines International Network. We supplemented this by searching societies that perform research

related to CKD. We performed a systematic search on the 14th September 2018, combining Medical Subject Headings (MeSH) terms and words related to CKD, ESKD, dialysis and RRT, limiting the results to Practice Guideline [publication type]. A full search strategy is shown in S1 Appendix. We updated the database search on 2nd August 2021. We used the same search method, except that we narrowed the searches to 2018 onwards.

## Selection of high-quality CPGs and data extraction

One reviewer (KS) screened the titles and abstracts of all records and discarded those that were duplicates or that were not pertinent for the study. Two reviewers (JC and KS) independently assessed the full text of potentially relevant guidelines and selected those that met the inclusion criteria. Disagreements were resolved through discussion. When consensus could not be reached, a third reviewer (IJO) was consulted.

We used the AGREE II instrument to identify high-quality CPGs from the previously selected guidelines [9]. The AGREE II instrument contains 23 items distributed along six domains of guideline development: Scope and Purpose, Stakeholder Involvement, Rigor of Development, Clarity of Presentation, Applicability, and Editorial Independence; and two overall assessments. Each item is rated on a seven-point Likert scale from 1 (strongly disagree) to 7 (strongly agree). A quality domain score between 0% and 100% is calculated based on this rating for each of the six domains [9].

Once high-quality CPGs were identified, one reviewer (KS) extracted the text, quality of evidence and strength of recommendations. This information was independently verified by the second reviewer (IJO). We did not assess the quality of the underlying evidence. Two reviewers assessed the content and variation across recommendations. The first reviewer (KS) identified topics covered by the recommendations and codified them. Codes then were reassessed through a comparison of each CPG recommendation to identify similarities and discrepancies. Finally, recommendations were grouped along the topics to analyse variation across CPGs. A second reviewer (IJO) then verified this information independently.

## Statistical analysis

We analysed the data using descriptive statistics: absolute and relative frequencies for categorical variables, measures of central tendency and dispersion for continuous variables. A quality score was calculated for each of the six AGREE II domains by summing up all the scores of the individual items in a domain and by scaling the total as a percentage of the maximum possible score for that domain [9]. The intraclass correlation coefficient (ICC) was used to measure inter-rater reliability in the AGREE II instrument scores. The degree of reliability was measured using the following ICC definitions: $\leq 0.50$ poor reliability; 0.51–0.75 moderate reliability; 0.76–0.90 good reliability; greater than 0.90 excellent reliability [14]. We conducted a narrative, descriptive synthesis to analyse the consistency between CPGs recommendations.

## Results

### Search and characteristics of included guidelines

The first main search retrieved a total of 2628 records, of which 1040 were duplicates. We excluded 1481 documents at the title and abstract screening, leaving 107 eligible records for full-text analysis. Eighty-nine records were excluded because they did not meet the inclusion criteria. The remaining 18 CPGs were assessed with the AGREE-II instrument. Seven were excluded because they did not obtain a minimum score of 60% in domains 3 (Rigour of development) or 6 (Editorial independence). References and reasons for the exclusion of CPGs

excluded in this phase are given in S1 Table. This left 11 high-quality CPGs for inclusion [15–25]. A PRISMA flow diagram of the selection process is shown in Fig 1.

The updated searches in August 2021 resulted in a total of 712 records being screened and assessed for eligibility. Four CPGs were evaluated with the AGREE-II instrument, and three were excluded because they did not obtain a minimum score of 60% in domains 3 (Rigour of development) or 6 (Editorial independence). References and reasons for the exclusion of CPGs excluded in this phase are given in S1 Table. One new high-quality CPG was included [26]. This selection process is also shown in Fig 1. In total, we screened 2101 records which resulted in 12 high-quality CPGs being included in the review [15–26].

Table 1 presents the characteristics of the included CPGs. Half of them were developed in Europe. Five CPGs (42%) were published within the last five years, and 10 (83%) were published in English. Three CPGs (25%) were focused on a specific population: people with diabetes mellitus and CKD [19], autosomal dominant polycystic kidney disease (ADPKD) [21], and older adults (>65 years old) with CKD [23]. The level of evidence and strength of recommendations was assessed through the Grades of Recommendation, Assessment, Development and Evaluation (GRADE) approach in 11 (92%) of the included CPGs. However, nine (75%) used a modified GRADE version. The remaining CPG (8%) used a generic grading system. Table 2 shows the differences between the systems of quality of evidence and strength of recommendation.

## Methodological quality

Tables 3 and S2 present AGREE II domain scores for each high-quality CPG. The median scores and the range for the domains were: scope and purpose 93% (72–100%); stakeholder

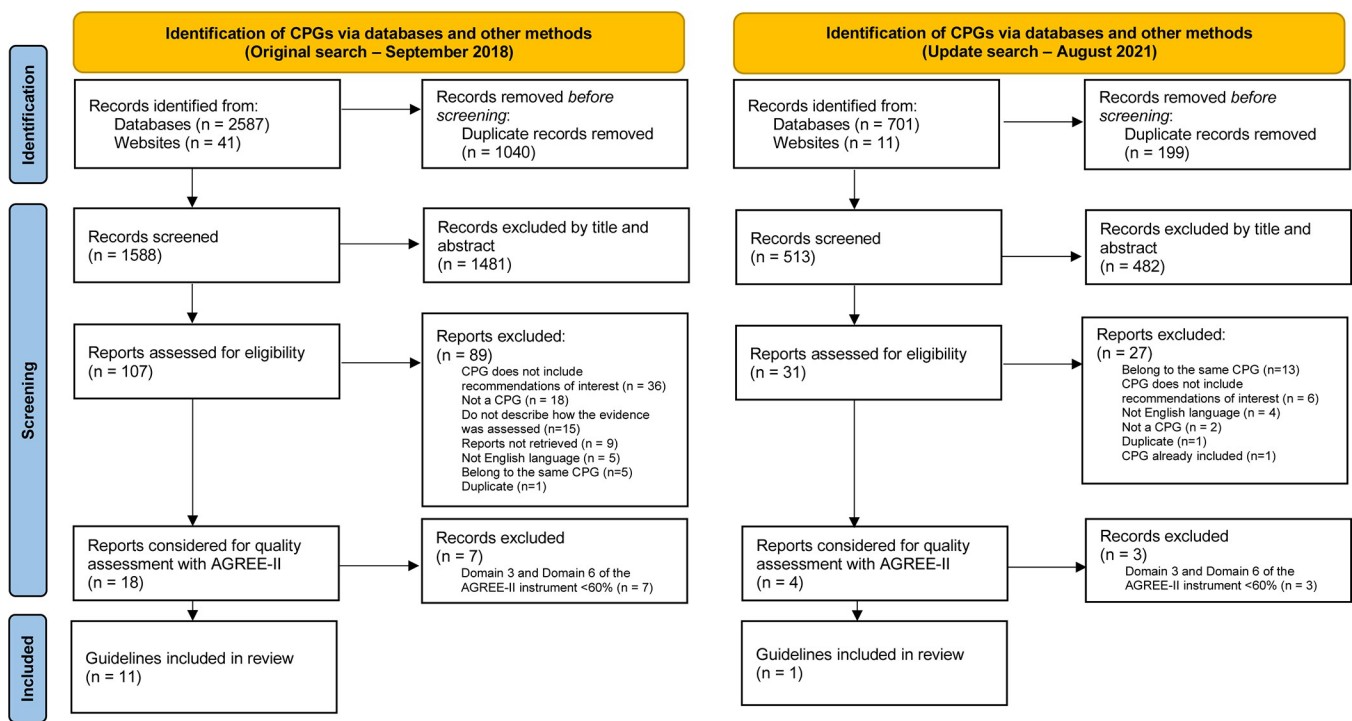

*From:* Page MJ, McKenzie JE, Bossuyt PM, Boutron I, Hoffmann TC, Mulrow CD, et al. The PRISMA 2020 statement: an updated guideline for reporting systematic reviews. BMJ 2021;372:n71. doi: 10.1136/bmj.n71. For more information, visit: http://www.prisma-statement.org

**Fig 1. PRISMA flow diagram.**

**Table 1. Characteristics of included high-quality clinical practice guidelines.**

| Organisation | Name | Country (Year) | Language | Target population | Theme (s) included | Levels of evidence Grade of recommendation |
|---|---|---|---|---|---|---|
| **Chile Ministry of Health** | Clinical guideline: peritoneal dialysis | Chile (2010) | Spanish | People with CKD treated with PD | Theme 2 & 3 | Unspecified* |
| **KDIGO** | KDIGO 2012 CPG for the Evaluation and Management of CKD | USA (2013) | English | People with CKD who are not on RRT | Theme 1 & 3 | Modified GRADE system |
| **UK Renal Association** | Planning, Initiating and Withdrawal of RRT | UK (2013) | English | People with CKD with established renal failure | Theme 1, 2 & 3 | Modified GRADE system |
| **Canadian Society of Nephrology** | CPG for timing the initiation of chronic dialysis | Canada (2014) | English | People with CKD for whom initiation of elective dialysis is planned | Theme 1 | Standard GRADE system |
| **ERBP** | CPG on the management of patients with diabetes and CKD stage 3b or higher | Europe (2015) | English | People with DM and CKD 3b or higher | Theme 1, 2 & 3 | Modified GRADE system |
| **National Kidney Foundation KDOQI** | KDOQI Clinical Practice Guideline for Haemodialysis Adequacy: 2015 update | US (2015) | English | People with CKD treated with, initiating, or planning to initiate maintenance HD | Theme 1 & 3 | Modified GRADE system |
| **KHA-CARI** | ADPKD Guideline: Management of End-Stage Kidney Disease | Australia (2015) | English | People with ADPKD | Theme 2 | Modified GRADE system |
| **Spain Ministry of Health** | CPG on detection and management of CKD | Spain (2016) | Spanish | People with CKD who are not on RRT | Theme 3 | Standard GRADE system |
| **ERBP** | CPG on the management of older patients with CKD stage 3b or higher | Europe (2016) | English | Older people (>65 years) with CKD | Theme 3 | Modified GRADE system |
| **UK Renal Association** | CPG Peritoneal Dialysis in Adults and Children | UK (2017) | English | People with CKD treated with PD | Theme 3 | Modified GRADE system |
| **NICE** | Renal replacement therapy and conservative management | UK (2018) | English | People with CKD 4 or 5 | Theme 1, 2 & 3 | Modified GRADE system |
| **International Society of Peritoneal Dialysis** | Prescribing High Quality Goal-Directed Peritoneal Dialysis | International (2020) | English | People with CKD treated with PD | Theme 2 | Modified GRADE system |

*Based on the quality of included studies and consensus amongst authors.

CKD: chronic kidney disease. PD: peritoneal dialysis. HD: haemodialysis. CPG: Clinical Practice Guideline. KDIGO: Kidney Disease: Improving Global Outcomes. GRADE: Grading of Recommendations Assessment, Development and Evaluation. ERBP: European Renal Best Practice. KDOQI: Kidney Disease Outcomes Quality Initiative. KHA-CARI: Kidney Health Australia—Caring for Australasians with Renal Impairment. NICE: National Institute for Health and Care Excellence. CEBM: Centre for Evidence-Based Medicine at Oxford. Theme 1: timing of initiating dialysis. Theme 2: selection of dialysis modality. Theme 3: Interventions to support the decision-making process.

involvement 80% (39–100%); rigour of development 82% (60–100%); clarity and presentation 99% (97–100%); applicability 57% (17–96%); and editorial independence 88% (63–100%). The intraclass coefficient (ICC) showed high values for all the CPGs, indicating good to excellent reliability.

## Variation across CPGs recommendations

While assessing and comparing the CPGs recommendations, we considered that CPGs had different target populations and primary objectives and did not cover all the aspects analysed in our systematic review (Tables 1 and S3). Table 4 shows the coded categories extracted from the text of high-quality CPGs recommendations. S4 Table shows the complete text of the recommendations.

Six of the included CPGs (50%) addressed recommendations related to the timing of initiating dialysis. We identified three topics across CPGs recommendations: 1) starting dialysis in

**Table 2. Levels of evidence and strength of recommendations.**

| ORGANISATION | LEVELS OF EVIDENCE | STRENGTH OF RECOMMENDATIONS |
|---|---|---|
| **Chile Ministry of Health** | **1** = Randomized controlled trial | **A** = Highly recommended, based in good-quality studies. |
| | **2** = Cohort, Case-control, no randomised trial | **B** = Based in moderate-quality studies. |
| | **3** = Descriptive studies | **C** = Exclusively based in experts' opinion or low-quality studies. |
| | **4** = Experts' opinion | **I** = Insufficient information to give a recommendation |
| **Canadian Society of Nephrology** | **Standard GRADE system** | **Standard GRADE system** |
| **Spain Ministry of Health** | **High** = We are very confident that the true effect lies close to that of the estimate of the effect | **Strong** = Most individuals in this situation would want the recommended course of action and only a |
| | **Moderate** = We are moderately confident in the effect estimate: The true effect is likely to be close to the estimate of the effect, but there is a possibility that it is substantially different | small proportion would not. Most patients should receive the recommended course of action. |
| | **Low** = Our confidence in the effect estimate is limited: The true effect may be substantially different from the estimate of the effect | **Weak** = The majority of individuals in this situation would want the suggested course of action, but many would not. Different choices will be appropriate for different patients. Each patient needs help to arrive at a management decision consistent with her or his values and preferences |
| | **Very low** = We have very little confidence in the effect estimate: The estimate of effect is very uncertain, and often will be far from the truth | |
| **KDIGO** | **Modified GRADE system** | **Modified GRADE system** |
| **ERBP** | **A** = High | **Level 1** = 'We recommend' (use the same implication as strong recommendation of the standard GRADE system) |
| **KDOQI** | **B** = Moderate | **Level 2** = 'We suggest' (use the same implication as weak recommendation of the standard GRADE system) |
| **KHA–CARI** | **C** = Low | **Not Graded / Practice Point** = Used to provide guidance based on common sense or where the topic does not allow adequate application of evidence. The ungraded recommendations are generally written as simple declarative statements but are not meant to be interpreted as being stronger recommendations than Level 1 or 2 recommendations. |
| **International Society of Peritoneal Dialysis** | **D** = Very low (same meaning as the standard GRADE system for the four categories) | |
| **UK Renal Association** | **Modified GRADE system** | **Modified GRADE system** |
| | **Grade A** = High-quality evidence that comes from consistent results from well-performed randomised controlled trials, or overwhelming evidence of some other sort such as well-executed observational studies with very strong effects | **Grade 1** = 'We recommend' (strong recommendation) |
| | **Grade B** = Moderate-quality evidence from randomised trials that suffer from serious flaws in conduct, inconsistency, indirectness, imprecise estimates, reporting bias, or some combination of these limitations, or from other study designs with special strength | **Grade 2** = 'We suggest' (weak recommendation) |
| | **Grade C** = Low-quality evidence from observational studies, or from controlled trials with several very serious limitations | Use wording to indicate the strength of each recommendation. When making a strong recommendation guideline authors are encouraged to use 'We recommend. . .' and when making a weak recommendation authors should use 'We suggest. . .' |
| | **Grade D** = Based only on case studies or expert opinion | |
| **NICE** | **Modified GRADE system** | **Modified GRADE system** |
| | **High** = further research is very unlikely to change our confidence in the estimate of effect | Use the wording of the recommendations: |
| | **Moderate** = further research is likely to have an important impact on our confidence in the estimate of effect and may change the estimate | Recommendations that should (or not should) be used = 'Offer' (or 'do not offer'), 'Advise', 'Ask about' or 'Commission' |
| | **Low** = further research is very likely to have an important impact on our confidence in the estimate of effect and is likely to change the estimate | Recommendation that could be used = 'Consider', 'Be aware of', 'Explore', 'Assess' or 'Think about' |
| | **Very Low** = any estimate of effect is very uncertain | Recommendations that must (or must not) be used = 'Must' (or 'must not') |

**Table 3. Domain scores of high-quality CPGs according to the AGREE II instrument.**

| Organization | Name | Domain scores (%) | | | | | | Intraclass correlation coefficient (95% CI) |
|---|---|---|---|---|---|---|---|---|
| | | Scope and Purpose | Stakeholder involvement | Rigour of development | Clarity and presentation | Applicability | Editorial independence | |
| Chile Ministry of Health | Clinical guideline: peritoneal dialysis | 97 | 58 | 68 | 100 | 71 | 96 | 0.94 (0.71–0.99) |
| KDIGO | KDIGO 2012 CPG for the Evaluation and Management of CKD | 100 | 92 | 90 | 100 | 46 | 92 | 0.98 (0.92–0.99) |
| UK Renal Association | Planning, Initiating and Withdrawal of Renal Replacement Therapy | 72 | 42 | 60 | 97 | 44 | 63 | 0.88 (0.46–0.98) |
| Canadian Society of Nephrology | CPG for timing the initiation of chronic dialysis | 100 | 75 | 89 | 100 | 96 | 92 | 0.97 (0.85–0.99) |
| ERBP | CPG on the management of patients with diabetes and CKD stage 3b or higher | 100 | 100 | 100 | 100 | 58 | 100 | 0.95 (0.73–0.99) |
| National Kidney Foundation KDOQI | KDOQI CPG for Haemodialysis Adequacy: 2015 update | 86 | 39 | 82 | 97 | 17 | 71 | 0.96 (0.81–0.99) |
| KHA—CARI | Autosomal Dominant Polycystic Kidney Disease Guideline: Management of ESKD | 81 | 89 | 77 | 100 | 35 | 83 | 0.89 (0.50–0.98) |
| Spain Ministry of Health | CPG on detection and management of CKD | 100 | 97 | 97 | 100 | 81 | 92 | 0.99 (0.97–0.99) |
| ERBP | CPG on the management of older patients with CKD stage 3b or higher | 100 | 94 | 100 | 100 | 58 | 96 | 0.99 (0.98–0.99) |
| UK Renal Association | CPG Peritoneal Dialysis in Adults and Children | 81 | 78 | 60 | 97 | 38 | 79 | 0.76 (0.11–0.96) |
| NICE | Renal replacement therapy and conservative management | 97 | 94 | 96 | 100 | 96 | 96 | 0.99 (0.98–0.99) |
| International Society of Peritoneal Dialysis | Prescribing High Quality Goal-Directed Peritoneal Dialysis | 100 | 97 | 64 | 100 | 42 | 92 | 0.80 (0.20–0.97) |
| | Median scores (range) | 93 (72–100) | 80 (39–100) | 82 (60–100) | 99 (97–100) | 57 (17–96) | 88 (63–100) | |

CPG: clinical practice guideline. KDIGO: Kidney Disease: Improving Global Outcomes. CKD: chronic kidney disease. ERBP: European Renal Best Practice. KDOQI: Kidney Disease Outcomes Quality Initiative. KHA-CARI: Kidney Health Australia—Caring for Australasians with Renal Impairment. ESKD: end-stage kidney disease. NICE: National Institute for Health and Care Excellence.

the presence of symptoms or signs related to CKD, 2) initiating dialysis at a specific starting point (eGFR) in the absence of symptoms, and 3) ensuring that the decision to start RRT is made jointly by the person and the healthcare team after a careful discussion. All six CPGs agreed on starting dialysis in the presence of symptoms or signs. Three CPGs considered this statement a strong recommendation, two CPGs considered it weak, and one CPG did not grade the recommendation. The quality of evidence reported varied from very low to high. Only two of the six CPGs recommended a specific eGFR of 5–7 ml/min/m$^2$ as a starting point in asymptomatic patients. The strength of the recommendation considered by these two CPGs differed from one to another. Likewise, only two CPGs strongly recommended that the

**Table 4. High-quality CPGs recommendations related to the process of starting dialysis.**

| CPGs RECOMMENDATIONS | HIGH-QUALITY CPGs INCLUDED IN THE STUDY | | | | | | | | | | | |
|---|---|---|---|---|---|---|---|---|---|---|---|---|
| | 1 [15] | 2 [16] | 3 [17] | 4 [18] | 5 [19] | 6 [20] | 7 [21] | 8 [22] | 9 [23] | 10 [24] | 11 [25] | 12 [26] |
| **TIMING OF DIALISYS INITIATION** | | | | | | | | | | | | |
| With the onset of signs or symptoms | | | | | | NG | | | | | | |
| Specific GFR (if symptoms are not present) | | | | | | | | | | | | |
| The decision to start should be based on joint discussion with the patient | | | | | | | | | | | | |
| **SELECTION OF DIALYSIS MODALITY** | | | | | | | | | | | | |
| Modality should be prescribed using shared decision making between person and care team | | | | | | | | | | | | NG |
| Offer modalities and ensure that the decisions are informed | | | | | | | | | | | | |
| No superiority within dialysis modalities for persons with ADPKD and DM | | | | | | | | | | | | |
| Encourage PD and home-dialysis use | | | | | | | | | | | | NG |
| Offer regular opportunities to review the original decision | | | | | | | | | | | | |
| **INTERVENTIONS TO SUPPORT THE DECISION-MAKING PROCESS ABOUT DIALYSIS MODALITY SELECTION** | | | | | | | | | | | | |
| Educational programs with information or counselling about RRT | | NG | ** | | | NG | | | | | | ** |
| Use of tools to predict CKD progression, prognosis, QoL, or mortality | | | | | | | | | | | | |
| Use of some type of patient decision-aid | | | | | | | | | | | * | * |
| | QUALITY OF EVIDENCE | | | | | | | | | | | |
| | High | | | Moderate | | | Low | | | Very low | | |
| **STRENGTH OF RECOMMENDATION** — Strong | | | | | | | | | | | | |
| Weak | | | | | | | | | | | | |

NG = not graded.

*Did not make a specific recommendation but offer information about this topic.

**Include recommendations about characteristics of the provided information and skills of the healthcare professional that provides it.

decision to start dialysis should be made jointly between the patient and the healthcare team (Tables 4 and S4).

Six of the included CPGs (50%) contained recommendations related to the selection of dialysis modality. We identified five different topics across guidelines making it difficult to assess the consistency across recommendations. Two CPGs explicitly mentioned the principle of shared decision-making and recommended ensuring a joint decision. One CPG made a strong recommendation, whereas the other one made a practice point. Two CPGs made strong recommendations about offering all dialysis modalities, ensuring informed decisions. Two CPGs with specific target populations (DM and ADPKD) strongly recommended that either treatment, HD or PD, be considered since there is an absence of evidence of the superiority of one modality over the other for these populations. Three CPGs made recommendations about encouraging the use of home dialysis (including PD) where possible. Two CPGs made a weak recommendation, whereas the other one did not grade it. Two CPGs recommended offering regular opportunities to review the original decision. One CPG made a weak recommendation, whereas the other made a strong one (Tables 4 and S4).

Nine of included CPGs (75%) had recommendations related to interventions to support the decision-making process about dialysis modality selection. Seven CPGs considered this statement a strong recommendation, and two CPGs did not grade the recommendation. Eight

CPGs agreed to recommend educational programs that contain information about the different RRT options. Two CPGs strongly recommended using tools to predict clinical outcomes to guide clinicians to find the best RRT option. Only one CPG considered using patient decision aids a strong recommendation based on moderate quality of evidence. (Tables 4 and S4).

## Discussion

### Summary of main findings

This systematic review of high-quality CPGs recommendations addressing the process of starting dialysis showed that, overall, there is general consistency in initiating dialysis late in the presence of symptoms or signs and offering patients education and information at the time of decision-making. Nevertheless, there is variability in how high-quality CPGs address the issue of dialysis modality selection and the use of decision tools other than education.

Across all CPGs, median domain scores at the AGREE II instrument were high, except for domain 5 (Applicability), where only four CPGs scored over 60%. This finding is consistent with what has already been described in studies that appraised the quality of CKD guidelines [27–30].

Few CPGs addressed when to start dialysis in asymptomatic patients. Although the recommended eGFR as the starting point was similar, the strength of the recommendation differed. This uncertainty is also seen in the current clinical practice, where there is variability in mean pre-dialysis eGFR among countries since a specific eGFR value for initiating dialysis without symptoms has not been established [31]. Using only symptom-based criteria might put at risk asymptomatic patients or those with subtle symptoms since there may be difficulties in identifying them.

Although we found consistency in content and strength of recommendation related to ensuring that the decision to start RRT is made jointly by the patient and the healthcare team, we think this topic is still rarely tackled. Planning starting dialysis often includes individualised discussions regarding patient values and preferences. With the increasing recognition of the importance of person-centred care, it could be expected that more high-quality CPGs would recommend a joint discussion with the patient about the decision to start dialysis. Similarly, shared decision-making is still a field that needs further discussion within high-quality CPGs since only two stated a specific recommendation about it. There is currently a need to provide more individualised care that incorporates the patient's goals and preferences. Shared decision-making relies on knowing and understanding the best available evidence about the risks and benefits across all available options while ensuring that the patient's values and preferences are considered [32]. The initiative Choosing Wisely published in 2012 a recommendation that dialysis should not be initiated without ensuring a shared decision-making process among patients, their families, and the healthcare team [33].

A variety of interventions have been designed to help shared decision-making to be implemented into clinical practice: 1) interventions targeting healthcare professionals; 2) interventions targeting patients; and 3) interventions targeting both [31]. In our systematic review, only the CPGs that focused on older people with CKD acknowledged recommendations advising on the use of prediction models or scores to predict progression or mortality in this population [23].

On the other hand, there was general consistency in high-quality CPGs in offering interventions targeting patients. Most of the guidelines recommended offering educational programs with information about the different RRT options. Few CPGs researched recommend the use of patient decision aids. The Spain Ministry of Health's CPG made a strong recommendation based on moderate evidence for using decision aids to help CKD patients make shared

decisions [22]. In contrast, the NICE CPG committee was unable to recommend that decision aids should be used because of the absence of evidence showing clinically important benefits [25]. The International Society of Peritoneal Dialysis guideline did not make a specific recommendation on this issue. Still, in the discussion section, they suggest that decision aids should be provided, including audio-visual as well as written material [26]. There is currently an increasing interest in using patient decision aids to support patients with CKD to make treatment modality decisions. There is some ongoing research that will offer more information about this topic [34, 35]. We did not find any high-quality CPGs that included recommendations about interventions targeting both patients and healthcare professionals.

## Quality of evidence and strength of recommendations

We observed considerable variations in the quality of evidence and the strength of recommendations across the CPGs that could be confusing since the implications of strong or weak recommendations are highly different [36]. Possible explanations about these differences have been previously described and included the year of CPG development, date of search by guideline development groups, differences in the methods used to identify and appraise evidence, or differences in the interpretation of the evidence [37]. Most of the CPGs included in our systematic review used a modified GRADE approach. Since GRADE aims to reduce confusion arising from multiple systems for grading evidence and recommendations, it would be reasonable to adopt the standard GRADE system. This could facilitate the comparison and understanding of the terminology.

## Comparison with the existing literature

Although we did not identify systematic reviews of CPGs addressing starting dialysis, we found a meta-synthesis of qualitative studies aimed to understand the process of decision-making in persons with CKD [38]. The review found that modality decisions are highly personal and strongly influenced by personal values. There is a need for planned and timely discussions about modalities in which home-based dialysis is presented as a viable option.

## Strength and limitations

We conducted comprehensive searches to identify relevant CPGs that addressed recommendations about the process of starting dialysis. We used a two-step process to identify high-quality CPGs. We used a validated tool (AGREE II) to assess the quality of included guidelines independently, and we obtained high ICC values meaning good reliability.

However, we recognise some limitations. We could have missed potentially relevant guidelines since we limited our search to CPGs published from 2008, and we set up a cut-off point of 60% in domains of the rigour of development and editorial independence for defining high-quality CPGs. Although we used the AGREE II instrument to assess the methodological quality of the CPGs, we did not evaluate the evidence underlying the recommendations. Although ICC values were usually high, using three or four appraisers would have potentially improve the reality of our assessment, especially for those CPGs with a wide confidence interval and for which the ICC was below 0.90.

## Implications for research

There is a dearth of evidence to inform current guidelines on when to start dialysis in asymptomatic patients and which modality to choose. We found variability in how CPGs reported the reasons and judgments behind the recommendations, suggesting there is variation in how

CPGs panels interpret and appraise the evidence. CPGs panels could consider adopting the GRADE Evidence to Decision framework for a structured approach in developing recommendations [39, 40]. This framework would facilitate the report of reasons and judgment that determine the direction and strength of recommendations across the different CPGs and allow comparison of recommendations from different CPGs panels.

### Implications for clinical practice

Since initiating and selecting a modality has as a centrepiece an individualised shared decision, offering individuals information about RRT options is probably not enough. High-quality CPGs should emphasise the best strategies and interventions to assess and incorporate the patient's values and preferences into clinical practice in their dialysis modality selection.

### Conclusions

High-quality CPGs related to the process of starting dialysis were consistent in initiating dialysis late, in the presence of symptoms or signs, and offering patients education and information at the time of decision-making. There was, however, variability in how high-quality CPGs address the issue of dialysis modality selection and the use of decision tools other than education. There is variation in the process used by different CPG panels to appraise the quality of evidence and grade the strength of recommendation. CPGs should improve their strategy on putting recommendations into practice and the quality of evidence to aid patients' decision-making.

### Supporting information

**S1 Appendix. Search strategy.**
(PDF)

**S1 Table. Excluded CPG with scores in domains 3 or 6 less than 60% in the AGREE-II instrument.**
(PDF)

**S2 Table. Complete scores of high-quality CPGs according to the AGREE II instrument.**
(PDF)

**S3 Table. Main objective of included high-quality CPGs.**
(PDF)

**S4 Table. Complete text of recommendations related to the start of dialysis.**
(PDF)

**S5 Table. PRISMA 2020 checklist.**
(PDF)

### Acknowledgments

Karla Salas-Gama is a PhD candidate at the Methodology of Biomedical Research and Public Health program, Universitat Autònoma de Barcelona, Spain.

### Author Contributions

**Conceptualization:** Karla Salas-Gama, Carl J. Heneghan.

**Data curation:** Karla Salas-Gama, Igho J. Onakpoya, Jorge Coronado Daza, Rafael Perera.

**Formal analysis:** Karla Salas-Gama, Igho J. Onakpoya, Jorge Coronado Daza, Rafael Perera.

**Investigation:** Karla Salas-Gama.

**Methodology:** Karla Salas-Gama, Igho J. Onakpoya, Jorge Coronado Daza, Rafael Perera.

**Supervision:** Igho J. Onakpoya, Carl J. Heneghan.

**Validation:** Igho J. Onakpoya, Jorge Coronado Daza, Carl J. Heneghan.

**Writing – original draft:** Karla Salas-Gama.

**Writing – review & editing:** Karla Salas-Gama, Igho J. Onakpoya, Jorge Coronado Daza, Carl J. Heneghan.

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
