## [Decision Letter · Decision Letter 0]

3 Feb 2022

PONE-D-21-37316Recommendations of high-quality clinical practice guidelines related to the process of starting dialysis: a systematic review.PLOS ONE

Dear Dr. Salas-Gama,

Thank you for submitting your manuscript to PLOS ONE. After careful consideration, we feel that it has merit but does not fully meet PLOS ONE’s publication criteria as it currently stands. Therefore, we invite you to submit a revised version of the manuscript that addresses the points raised during the review process.

We look forward to receiving your revised manuscript.

Kind regards,

Pierre Delanaye

Academic Editor

PLOS ONE

Journal Requirements:

I have read the journal's policy and the authors of this manuscript have the following competing interests: 

IJO and CJH have held grant funding from the NIHR Evidence Synthesis Working Group (ESWG Grant no: 390). CJH has received expenses and fees for his media work (including payments from BBC Radio 4 Inside Health). He has received expenses from the WHO, FDA, and holds grant funding from the NIHR, the NIHR School of Primary Care Research, the NIHR BRC Oxford and the WHO. He has received financial remuneration from an asbestos case and given free legal advice on mesh cases. He has also received income from the publication of a series of toolkit books published by Blackwells. On occasion, he receives expenses for teaching EBM and is also paid for his GP work in NHS out of hours (contract with Oxford Health NHS Foundation Trust). RP hold grant funding from the NIHR Programme of Applied Research. He leads a programme looking at how general practitioners manage chronic kidney disease and chronic heart failure. 

KS and JC have no interests to declare. 

3. Please ensure that you include a title page within your main document. You should list all authors and all affiliations as per our author instructions and clearly indicate the corresponding author.

Additional Editor Comments:

Very good article. Only minor comments need to be addressed.

Reviewers' comments:

Reviewer's Responses to Questions

**Comments to the Author**

1. Is the manuscript technically sound, and do the data support the conclusions?

Reviewer #1: Yes

Reviewer #2: Yes

Reviewer #3: Yes

2. Has the statistical analysis been performed appropriately and rigorously? 

Reviewer #1: Yes

Reviewer #2: Yes

Reviewer #3: Yes

3. Have the authors made all data underlying the findings in their manuscript fully available?

Reviewer #1: Yes

Reviewer #2: Yes

Reviewer #3: Yes

4. Is the manuscript presented in an intelligible fashion and written in standard English?

Reviewer #1: Yes

Reviewer #2: Yes

Reviewer #3: Yes

5. Review Comments to the Author

Reviewer #1: This paper describes an AGREE evaluation of guidelines on starting renal replacement therapy in patients with ESKD.

The paper is well writen, and the methodology used seems to be sound.

Conclusions are in line with the findings.

Reviewer #2: This a very good quality manuscript overall. Suggestions and questions have been left as commentaries in the left margin of the attached Word document. Please disregard the disrupted layout of the tables in the document as these have resulted from the exportation of the original pdf to a Word file in order to facilitate the reviewing process.

Reviewer #3: A well written narrative of review of guidelines concerning dialysis initiation.

In the introduction, I suggest to not overinterprated the paper of Korevaar and coll (ref4) since this RCT was not completed.

"A randomised controlled trial [4] compared starting dialysis with HD vs 49 PD showed no differences in mortality at two years. The results suggested a hypothesis of 50 equivalence between PD and HD."

No further comments.

6. PLOS authors have the option to publish the peer review history of their article (what does this mean?). If published, this will include your full peer review and any attached files.

Reviewer #1: No

Reviewer #2: No

Reviewer #3: **Yes: **Cécile Couchoud

---

## [Author Response · Author response to Decision Letter 0]

15 Mar 2022

Dear Reviewers,

Thank you for revising our manuscript. We have incorporated all of your suggestion into our manuscript. They were very helpful. Thank you very much.

---

## [Editor Report · Decision Letter 1]

16 Mar 2022

Recommendations of high-quality clinical practice guidelines related to the process of starting dialysis: a systematic review.

PONE-D-21-37316R1

Dear Dr. Salas-Gama,

We’re pleased to inform you that your manuscript has been judged scientifically suitable for publication and will be formally accepted for publication once it meets all outstanding technical requirements.

Kind regards,

Pierre Delanaye

Academic Editor

PLOS ONE
---

## [Editor Report · Acceptance letter]

1 Apr 2022

PONE-D-21-37316R1 

Recommendations of high-quality clinical practice guidelines related to the process of starting dialysis: a systematic review. 

Dear Dr. Salas-Gama:

I'm pleased to inform you that your manuscript has been deemed suitable for publication in PLOS ONE. Congratulations! Your manuscript is now with our production department. 

Kind regards, 

on behalf of

Professor Pierre Delanaye 

Academic Editor

PLOS ONE